# Antiangiogenic Compound Axitinib Demonstrates Low Toxicity and Antitumoral Effects against Medulloblastoma

**DOI:** 10.3390/cancers14010070

**Published:** 2021-12-24

**Authors:** Marina Pagnuzzi-Boncompagni, Vincent Picco, Valérie Vial, Victor Planas-Bielsa, Ashaina Vandenberghe, Thomas Daubon, Marie-Alix Derieppe, Christopher Montemagno, Jérôme Durivault, Renaud Grépin, Sonia Martial, Jérôme Doyen, Julie Gavard, Gilles Pagès

**Affiliations:** 1Biomedical Department, Centre Scientifique de Monaco, 98000 Monaco, Monaco; mpagnuzzi@centrescientifique.mc (M.P.-B.); vial@centrescientifique.mc (V.V.); ashaina.vandenberghe@gmail.com (A.V.); cmontemagno@centrescientifique.mc (C.M.); jdurivault@centrescientifique.mc (J.D.); rgrepin@centrescientifique.mc (R.G.); 2Polar Biology Department, Centre Scientifique de Monaco, 98000 Monaco, Monaco; vplanas@centrescientifique.mc; 3Institut de Biochimie et Génétique Cellulaires (IBGC), CNRS, University of Bordeaux, UMR 5095, 33000 Bordeaux, France; thomas.daubon@u-bordeaux.fr; 4Animalerie Mutualisée, Service Commun des Animaleries, University of Bordeaux, 33600 Pessac, France; marie-alix.derieppe@u-bordeaux.fr; 5Centre Antoine Lacassagne, Institute for Research on Cancer and Aging of Nice (IRCAN), University Nice Cote d’Azur, CNRS UMR 7284, INSERM U1081, 06189 Nice, France; sonia.martial@univ-cotedazur.fr; 6Department of Radiation Oncology, Centre Antoine-Lacassagne, University of Côte d’Azur, Fédération Claude Lalanne, 06189 Nice, France; Jerome.DOYEN@nice.unicancer.fr; 7Team SOAP, CRCINA, INSERM, CNRS, Université de Nantes, 44000 Nantes, France; julie.gavard@univ-nantes.fr; 8Integrated Center of Oncology, 44800 St. Herblain, France

**Keywords:** pediatric brain cancer, medulloblastoma, targeted therapy, angiogenesis

## Abstract

**Simple Summary:**

Medulloblastoma is the most frequent pediatric brain cancer. Despite great improvements in the treatment of this disease over the last decades, survivors are subject to debilitating adverse effects that strongly impair their quality of life. There is an urgent need to find efficient anticancer therapies with fewer toxic effects. In this study, we suggest that an FDA- and EMA-approved antiangiogenic compound named axitinib may display effective antitumoral effects and low toxicity towards children as compared to a reference treatment currently used in clinical protocols. We also show that this compound can enter the brain compartment and exert antitumoral effects in vivo. Our study paves the way towards a clinical trial of repurposing axitinib to a pediatric brain cancer indication.

**Abstract:**

Background: Despite the improvement of medulloblastoma (MB) treatments, survivors face severe long-term adverse effects and associated morbidity following multimodal treatments. Moreover, relapses are fatal within a few months. Therefore, chemotherapies inducing fewer adverse effects and/or improving survival at relapse are key for MB patients. Our purpose was to evaluate the last-generation antiangiogenic drugs for their relevance in the therapeutic arsenal of MB. Methods: We screened three EMA- and FDA-approved antiangiogenic compounds (axitinib, cabozantinib and sunitinib) for their ability to reduce cell viability of five MB cell lines and their low toxicity towards two normal cell lines in vitro. Based on this screening, single-agent and combination therapies were designed for in vivo validation. Results: Axitinib, cabozantinib and sunitinib decreased viability of all the tested tumor cells. Although sunitinib was the most efficient in tumor cells, it also impacted normal cells. Therefore, axitinib showed the highest selectivity index for MB cells as compared to normal cells. The compound did not lead to acute toxicity in juvenile rats and crossed the blood–brain barrier. Moreover, axitinib efficiently reduced the growth rate of experimental brain tumors. Analysis of public databases showed that high expression of axitinib targets correlates with poor prognosis. Conclusion: Our results suggest that axitinib is a compelling candidate for MB treatment.

## 1. Introduction

Cancer is the second cause of mortality during childhood in high-income countries after accidental death. Though high during the last decades, the decreased rate of child cancer mortality tends to reach a plateau [1]. This suggests that the current anticancer drugs are reaching maximum optimization. New improvements in childhood cancer care will therefore need the development of new therapeutic approaches.

Cancers of the central nervous system are the second most prevalent childhood tumors after hematologic cancer [2]. Medulloblastoma (MB) is the most prevalent brain cancer in children and infants and accounts for 15–20% of childhood nervous system tumors [3]. The current therapeutic approach comprises surgical removal of the tumor, craniospinal radiation therapy (RT) and chemotherapy. It is based on the subdivision of patients into standard or high-risk groups based on the presence of metastases, age, extent of postsurgical residual disease and histology [4]. Patients belonging to the standard risk group receive lower doses of radio- and chemotherapy to limit the deleterious effects of treatments as much as possible.

Although optimization of MB treatment increased the long-term survival and decreased recurrence, the rate of deleterious late effects (occurring more than 5 years after diagnosis) also significantly increased from the 1970s to the 1990s [5]. The severe late outcomes include occurrence of a secondary neoplasm, severe psychological disorders and cardiac toxicity [5,6,7]. These side effects are due to the introduction of adjuvant systemic chemotherapies in the treatment of MB during the 1980s [5]. Therefore, finding new treatments that allow the same or increased survival while reducing adverse effects is an urgent need. 

Genetic characterization of the disease has led to the classification of MBs into subtypes: the wingless (Wnt), the sonic Hedgehog (Shh) and the more similar though molecularly distinguishable groups 3 and 4 [2]. MB subgrouping could orient the therapeutic approach. Indeed, targeted therapies may present fewer off-target effects compared to classical genotoxic chemotherapies and therefore represent a promising approach to increase tolerability of the treatments and reduce deleterious side effects [8]. The Hedgehog (Hh) signaling pathway is inappropriately activated in the Shh genetic subgroup of MB, especially through constitutive activation of the serpentine receptor Smoothened. Therefore, specific inhibitors of this receptor were developed, including sonidegib (LDE225) and vismodegib (GDC-0449). Unfortunately, these treatments showed limited response rates, potentially because of a specific mutation of Smoothened associated with permanent defects in bone growth [9,10,11].

Increased angiogenesis is associated with the most aggressive MBs (group 3) [12]. Moreover, direct cytotoxic effects towards the tumor cells aberrantly expressing the targets of antiangiogenic drugs have been observed [13]. Hence, antiangiogenic treatments might be of interest for MBs. The VEGF-targeting monoclonal antibody bevacizumab was used in combination with nontargeted chemotherapeutic agents for the treatment of pediatric patients with solid tumors, including some brain tumors [14,15,16,17,18]. Despite good tolerability of bevacizumab, no modification of event-free, progression-free or overall survival was observed. A previous work from our laboratory showed that multitarget antiangiogenic tyrosine kinase inhibitors (TKi) are directly toxic for cancer cells in addition to their in vivo antiangiogenic activity [13]. Amongst these, the TKi sunitinib has been tested in a pediatric setup, but a lack of benefits and, more importantly, an increase in the occurrence of adverse events were observed [19]. The efficacy of axitinib, another multitarget antiangiogenic TKi, for treating children with solid tumors has also been evaluated [20]. A phase I study determined the maximum tolerated and recommended dose in children presenting with refractory or recurrent tumors (2.4 mg/m^2^). Finally, cabozantinib is currently under clinical investigation regarding treatment of pediatric cancers, but no result is available, to our knowledge [21,22]. 

Our study aimed at determining the efficacy of antiangiogenic treatments on MB cells of the different genetic subgroups in vitro and on experimental tumors generated with Shh and Group3 cells. We particularly focused on tumors from the most aggressive group 3. We found that the FDA- and EMA-approved drugs sunitinib, cabozantinib and axitinib kill MB cells in vitro. However, axitinib presented the best selectivity towards cancer cells when compared to primary normal cells. No acute toxicity of this compound towards young and growing rats was observed [23]. Moreover, axitinib was detected in the brain of the animals and permeabilized an in vitro blood–brain barrier (BBB) model. Axitinib also displayed in vivo efficacy, both in subcutaneous and intracranial MB xenografts. Our results suggest that axitinib represents an option for patients in therapeutic deadlock. 

## 2. Materials and Methods

### 2.1. Cell Lines

DAOY cells (ATCC, HTB-186) were maintained in the MEM α supplemented with 7.5% fetal calf serum (FCS) (Dominique Dutscher SAS, Bernolsheim, France), 0.25% Glutamax, 1% NEAA and 0.1% sodium pyruvate (Thermo Fisher Scientific Inc., Waltham, MA, USA). D283-Med (ATCC, HTB-185) and D341-Med cells (ATCC, HTB-187) were cultured in the MEM (Thermo Fisher Scientific Inc., Waltham, MA, USA) supplemented with 15% fetal calf serum (Dominique Dutscher SAS, Bernolsheim, France). CHLA-01-Med cells (ATCC, CRL-3021) were cultured in DMEM/F12 (Thermo Fisher Scientific Inc., Waltham, MA, USA) supplemented with 2% B-27 (Fisher Scientific, Hampton, NH, USA), 20 ng/mL EGF and 20 ng/mL basiFGF (Sigma-Aldrich, St. Louis, MO, USA). HD-MB03 cells (DMSZ, ACC 740) were maintained in the RPMI medium (Thermo Fisher Scientific Inc., Waltham, MA, USA) supplemented with 7.5% FCS. D458 Med (Cellosaurus, CVCL_1161) were cultured in the Improved MEM (Thermo Fisher Scientific Inc., Waltham, MA, USA) with 7.5% FCS and 0.25% Glutamax. Human dermal fibroblasts (HDF) (Sigma-Aldrich 106-05N) were cultured in a fibroblast growth medium (Sigma-Aldrich). C8-D1A cells (ATCC CRL-2541) were maintained in DMEM (Thermo Fisher Scientific Inc., Waltham, MA, USA) supplemented with 7.5% SVF. Immortalized human cerebral microvascular endothelial cells (hCMEC/D3) and human umbilical vein endothelial cells (HUVEC) were maintained in Endothelial Basal Medium-2 (EBM-2, Lonza group Ltd., Switzerland) containing 5% fetal bovine serum (FBS Serum Gold, PAA Laboratories), 1% penicillin/streptomycin (P/S), HEPES and a chemically defined lipid concentrate (Thermo Fisher Scientific Inc., Waltham, MA, USA), hydrocortisone (1.4 mM), ascorbic acid (5 mg/mL) and the basic fibroblast growth factor (1 ng/mL; Sigma) as described by Weksler et al. [24]. DAOY, HD-MB03 and D458 cell lines are of human origin and were obtained from Dr. Celio Pouponnot’s laboratory (Institut Curie, Paris, France). D283-Med, D341-Med, CHLA-01-Med cells are also of human origin and were purchased from ATCC. C8-D1A cells are of murine origin and were purchased from ATCC. HDF cells were purchased from Sigma-Aldrich. HUVEC cells were purchased from Lonza. The absence of mycoplasma was verified on a bimonthly basis using a PlasmoTest kit (Invivogen, cat. code rep-pt1). DAOY cells belong to the Shh genetic subtype, CHLA-01-Med—to group 4, D341-Med, HD-MB03 and D283-Med—to group 3.

### 2.2. Lentiviral Infections

Lentiviral particles were prepared according to the standard protocol as described previously [25]. Briefly, lentiviral vectors pLenti-CMV-V5-Luc (plasmid 21474, Addgene, Watertown, MA, USA) or pLV-mCherry (plasmid 36084, Addgene, Watertown, MA, USA) were co-transfected with lentivirus packaging vectors psPAX2 (plasmid 12260, Addgene, Watertown, MA, USA) and pMD2.G (plasmid 12259, Addgene, Watertown, MA, USA) into HEK293T cells through PEI transfection. DAOY and HD-MB03 cells (1 and 5 million, respectively) were seeded in 100 mm diameter culture dishes and incubated with a viral supernatant (1:10 V:V) at 37 °C overnight. The medium was then replaced with a fresh medium. Cells stably infected with pLV-mCherry were selected for 7 days with 1 µg/mL puromycin (Thermo Fisher Scientific Inc., Waltham, MA, USA). 

### 2.3. MTT Proliferation Assay, Cell Viability and Cumulative Population Doubling Assays

HDF (3000 cells), C8-D1A (5000 cells), D458 (5000 cells), DAOY (2000 cells), HD-MB03 (10,000 cells) and CHLA-01-Med (10,000 cells) were seeded at an approximate confluency of 50% in 96-well plates (Corning Inc., Corning, NY, USA) in 100 µL medium per well. A range from 50 nM to 150 µM of every drug was tested. The effects of the antiangiogenic compounds were measured using a 3-[4,5-dimethylthiazol-2yl]-diphenyltetrazolium bromide (MTT) colorimetric assay (Sigma, Lyon, France) according to the manufacturer’s instructions. When applicable, IC50 was calculated using the nonlinear regression method of the Prism 5 software (Graphpad Software Inc., San Diego, CA, USA). Alternatively, hCMEC/D3 [23] or HUVEC cells were plated at 5000 cells/well and incubated 24 h later for 2 and 3 days with 1, 5 or 25 µM of axitinib. An MTT assay was conducted as described above to assess the fitness of the cells.

Cell viability assays were conducted using a propidium iodine incorporation technique with an ADAM-sHIT automated cell counter (NanoEnTek, Gyeonggi-do, Korea) according to the manufacturer’s protocol. Briefly, HDF (10,000 cells), C8-D1A (15,000 cells), D458 (15,000 cells), DAOY (10,000 cells), HD-MB03 (30,000 cells), D283 (30,000 cells) and CHLA-01-Med (30,000 cells) cultured in 12-well plates were treated for 48 h with 5 µM axitinib (S1005, Selleck Chemicals, Houston, TX, USA), cabozantinib (S1119, Selleck Chemicals, Houston, TX, USA), sunitinib (S7781, Selleck Chemicals, Houston, TX, USA) or a combination of etoposide (S1225, Selleck Chemicals, Houston, TX, USA) and carboplatin (S1215, Selleck Chemicals, Houston, TX, USA) at 1 µM and 1.6 µM, respectively, before the measurement.

DAOY, D458, D283, C8-D1A and HDF (30,000 cells per well) and HD-MB03 (50,000 cells per well) were seeded in 24-well plates for proliferation assays. The cells were counted the next day (time 0) and after 48 and 120 h. The media were replaced every 48 h to ensure constant concentrations of the drugs. The proliferation index was determined by dividing the number of cells at a given time by the initial number of cells. 

Cumulative population doubling (CPD) was assessed as previously described [25]. Briefly, depending on the cell line, a variable number of cells corresponding to about 30% confluency were seeded in triplicate in 60 mm diameter Petri dishes. The treatments started the next day and the media were replaced every 48 h to ensure constant concentrations of the drugs. The cells were harvested when the confluency reached about 90%, counted and plated again at a dilution of 1/20. CPD was calculated as previously described [25]. 

### 2.4. Coculture and FACS Experiments

C8-D1A cells were stained with the green fluorescent probe neuro-DiO (cat. number 60015, Biotium, Fremont, CA, USA) according to the manufacturer’s protocol. C8-D1A-neuroDio cells (150,000) and HD-MB03-mCherry or DAOY-mCherry cells (50,000) were plated in 12-well plates. The cells were exposed to axitinib (5 µm) or Eto/Carbo (1 µM/1.6 µM) treatment for 3 days, the medium was washed and the cells were left to recover for 3 days without treatment. Fluorescence images of the cells were taken with a DMI400 inverted microscope (Leica Microsystemes SAS, Nanterre, France) equipped with a 40× objective and a Zyla 5.5 camera (Andor Technologies, Belfast, UK). The cells were then trypsinized and the number of both cell types was quantified using a Melody FACS (BD Biosciences, Frankton Lake, NJ, USA) with a 488 nm laser beam. Analysis of the FACS data was conducted with the Flowjo software (Tree Star Inc., Ashland, OR, USA).

### 2.5. Spheroid Assays

Two thousand cells were seeded in ultralow-adhesion 96-well plates (Corning Inc., Corning, NY, USA). After 4 days, they were transferred in DMEM with 7.5% FCS supplemented with 5 μM axitinib or 1 µM etoposide and cultured for 8 days. Pictures were taken with an AMG Evos microscope 40× objective (Thermo Fisher Scientific Inc, Waltham, MA, USA) and the spheroid areas were measured using the ImageJ software (NIH, Bethesda, MD, USA).

### 2.6. Subcutaneous Xenografts

For subcutaneous xenograft experiments, tumor cells expressing the luciferase (350,000 HDMB-Luc cells and 1.10^6^ DAOY-Luc cells) were resuspended in 200 µL of 5 µg/mL Matrigel (Corning Inc., Corning, NY, USA) and injected subcutaneously in the flank of 5-week-old Rj:NMRI-Foxn1 nude (nu/nu) female mice (Janvier Labs, Le Genest-Saint-Isle, France). Saline solution (NaCl 0.9M; 100 µL) containing 30 mg/mL D-Luciferin (Perkin Elmer, Wellesley, MA, USA) was injected intraperitoneally in the animals and the bioluminescence was quantified using an In Vivo Imaging System (Perkin Elmer, Wellesley, MA, USA) according to the manufacturer’s instructions. Tumor volume (V = L × l2 × 0.52) was determined with a caliper and the endpoint was determined as a tumor volume of 1000 mm^3^. Axitinib (50 mg/kg), etoposide (30 mg/kg) (Selleck Chemicals, Houston, TX, USA) and a mix of 25 mg/kg axitinib and 15 mg/kg etoposide resuspended in 200 µL of an aqueous solution of 0.5% carboxymethylcellulose and 0.4% Tween 80 were administered by oral gavage three times a week. The mice were sacrificed when the tumor reached 1000 mm^3^. The tumors that did not reach the size of 500 mm^3^ were considered to be outliers and were not considered during the analysis of the experiments. No statistical methods were used to predetermine the sample size of the experiments. Animal facility availability and cost were used to determine sample sizes. Two independent cohorts of five mice per group were used for the tumor growth experiment with HD-MB03 cells and one cohort of five mice per group was used for the experiment with DAOY cells. Animal experiments were carried out in strict accordance with the recommendations in the Guide for the Care and Use of Laboratory Animals. Our experiments were approved by Direction de l’Action Sanitaire et Sociale of the Principality of Monaco and the ethics committee of Centre Scientifique de Monaco.

### 2.7. Intracranial Tumor Xenografts and Metronomic Administration of the Treatments

HD-MB03-Luc spheroids were stereotaxically implanted into the brains of 9-week-old Rj:NMRI-Foxn1 nude (nu/nu) female mice (Janvier Labs, Le Genest-Saint-Isle, France). Briefly, MB spheroids were generated with 2500 cells grown for 48 h in ultralow-adhesion spheroid 96-well plates (Corning, Corning, NY, USA) and three spheroids per mouse were implanted into the left cerebellar hemisphere (2 mm posterior, 1.5 mm left of the lambda point and 2.5 mm deep) using a Hamilton syringe fitted with a needle (Hamilton, Bonaduz, Switzerland) and following the procedure already described [26]. The mice were treated with 50 mg/kg axitinib, 30 mg/kg etoposide (Selleck Chemicals, Houston, TX, USA) or 25 mg/kg axitinib and 15 mg/kg etoposide resuspended in 200 µL of an aqueous solution of 0.5% carboxymethylcellulose and 0.4% Tween 80 administered by oral gavage on a metronomic scheme (five times a week at fixed time). Oral gavage with a vehicle solution was used as the control. Mouse survival was based on the presence of neuropathological symptoms including gait defects and a loss of weight over 10% as endpoints. A minimum of five mice per group was chosen to yield sufficient statistical power (*p* < 0.05, one-way ANOVA). These experiments were carried out in strict accordance with the recommendations in the Guide for the Care and Use of Laboratory Animals. Our experiments were approved by Direction de l’Action Sanitaire et Sociale of the Principality of Monaco and the ethics committee of Centre Scientifique de Monaco.

### 2.8. Tumor Growth Analysis

The growth rate of the tumors generated with HD-MB03 cells was determined using a linear regression method applied to the growth curves between the sizes of 400 mm^3^ up to the endpoint using Prism 5 (Graphpad Software Inc., San Diego, CA, USA). The growth of the tumors generated with DAOY cells could not be fitted to a linear curve and was therefore modeled according to a lognormal model (refer to the Supplementary Methods for details).

### 2.9. Histopathology and Automated Image Analysis

Tumor samples were recovered from the animals and embedded in the OCT compound according to the manufacturer’s protocol (Thermo Fisher Scientific Inc., Waltham, MA, USA). After that, 5 µm thin sections were prepared with a cryostat (Leica Microsystems, Wetzlar, Germany). Incubation was carried out with the following antibodies diluted at 1:1000 in TBS supplemented with 1% horse serum and 1% BSA for 20 min at room temperature: anti-Ki67 (cat. number ab16667, Abcam, Cambridge, UK), anti-CD31 (cat. number 550274, BD Biosciences, Frankton Lake, NJ, USA) or anti-αSMA (cat. number A2547, Merck Millipore, Burlington, MA, USA). The preparations were then washed with TBS–0.025% Triton, incubated with anti-rabbit Alexa488 and anti-mouse Alex-555-coupled secondary antibodies (Cell Signaling Technology, Danvers, MA, USA), washed with TBS–0.025% Triton, and the nuclei were counterstained with Hoechst33342 (Thermo Fisher Scientific, Waltham, MA, USA). Fluorescence images of the cells were taken with a DMI400 inverted microscope equipped with a 40× objective (Leica Microsystemes, Nanterre, France) and a Zyla 5.5 camera (Andor Technologies, Belfast, UK) powered by the Micromanager software [27]. At least one cross-section of nonoverlapping images of each tumor was acquired. All the images were then quantified with the CellProfiler 3.0 software [28].

### 2.10. Juvenile Rats Toxicity Experiment and Measurement of the Axitinib Concentration in the Brain

Toxicity experiments were carried out on 20-day-old RjHan:WI (Wistar) rats (10 males and 10 females) treated with 50 mg/kg axitinib resuspended in 100 µL of a 0.5% carboxymethylcellulose, 0.4% Tween 80 aqueous solution on a daily basis by oral gavage. All the experiments were serviced by Janvier Labs Company (Saint Berthevin, France) and conducted at a Janvier Labs facility in strict accordance with the recommendations in the Guide for the Care and Use of Laboratory Animals. The experiments were approved by Comité National Institutionnel d’Ethique pour l’Animal de Laboratoire (France). The rats were bled and the brains were recovered and flash-frozen at the end of the experiment. These samples were then sent to TechMedIll (Illkirch, France) to perform an LC/MS-based measurement of the axitinib concentration.

### 2.11. BBB Permeability Assay

This assay was conducted as previously described [29]. Briefly, hCMEC/D3 or HuVEC cells were plated at 250,000 cells/insert and incubated 24 h later for 3 days with 0.5 or 1 µM axitinib. Permeability was measured by the passage of 40 kDa FITC-Dextran through the inserts. Fluorescence intensity measures were normalized to the DMSO control.

### 2.12. Antibody Array Screenings

The cells were seeded in duplicate in six-well plates and treated with 10 µM axitinib or an equivalent amount of DMSO for 24 h. Lysis and antibody array hybridization were conducted as recommended by the manufacturers. Antibody arrays used in this study were as follows: RTK phosphorylation array 1 (cat. number 126AAH-PRTK-1-8, Raybiotech), Proteome Profiler Human Phospho-RTK and Phospho-Kinase Array Kit (ARY001B and ARY003B, R&D System).

### 2.13. Statistical Analyses

All the experiments were carried out in technical triplicates and repeated at least twice (the number of replicates is provided in the figure legends). One-way ANOVA tests were used to measure significance between several experimental groups. Student’s *t*-tests were used to measure significance between the control and an experimental group when indicated in the figure’s legend. The data were analyzed with the Prism 5 software (Graphpad Software Inc., San Diego, CA, USA). The tests were performed with a nominal significant level of 0.05; * *p* < 0.05, ** *p* < 0.01, and ** *p* < 0.001. The results are shown as the means ± SEM.

## 3. Results

### 3.1. Antiangiogenic Compounds Are Effective against MB Cells in 2D and 3D Culture Conditions

In addition to their effects on blood vessels, antiangiogenic compounds often present with cytotoxicity towards tumor cells [13]. Hence, we compared the effect of three antiangiogenic compounds (axitinib, cabozantinib and sunitinib) with that of the etoposide/carboplatin combination (1/1.6 ratio respectively, as in the pharmaceutical formulation), a chemotherapy used in clinic to treat high-risk MBs [30]. In parallel, the effects of the compounds towards nontumor cells were evaluated. Murine primary astrocytes (C8-D1A) were chosen based on their proliferative capacities and their neural lineage origin. Human primary dermal fibroblasts (HDF) were chosen based on their proliferative capacities and their human origin. The IC50 of each compound was determined on these normal cells and on MB cell lines (Table 1 and Appendix A). We showed that axitinib impacts group 3 (D283, D458 and HD-MB03) and putative Shh group (DAOY) MB cell lines [31] in a range of concentrations comparable to the reference treatment (mean IC50 1.1 ± 0.9 µM and 0.7 ± 0.3 µM, respectively) (Table 1 and Appendix A). We then calculated the selectivity index (SI) for each compound (Table 2) [32]. SI was determined as the ratio of the IC50 in normal cells and the IC50 in MB cell lines. An SI value above 1 indicates that a given compound is more efficient in tumor cells than in normal cells. An SI value higher than 5 is generally considered relevant for efficacy and low toxicity of a compound [32]. IC50 of axitinib in astrocytes and fibroblasts being extremely high, this calculation resulted in a much higher SI of axitinib as compared to all the other treatments, including the reference treatment (Table 2). Axitinib was therefore the only antiangiogenic drug that displayed a selective effect towards MB versus nontumor cells.

We also assessed the respective efficiency of etoposide and carboplatin in anticipation of in vivo experiments and a potential clinical trial where axitinib would have to be used in combination with another single-agent therapy. IC50 of carboplatin was above 20 µM for DAOY, HD-MB03 and D458 MB cell lines as well as for HDF and C8-D1A normal cells. Moreover, the effect of the etoposide/carboplatin combination was comparable to that of etoposide alone (Table 3). We therefore chose to discard carboplatin and use etoposide alone in the subsequent experiments.

We next evaluated the effect of each compound on the viability and proliferation of all the five MB cell lines and of the nontumor primary cells used in the previous experiments (Figure 1 and Appendix A). Axitinib, cabozantinib and sunitinib decreased the viability of most MB cells (Figure 1A,B). Furthermore, axitinib, cabozantinib and etoposide treatments had no effect on the viability of HDF and C8-D1A normal cells while sunitinib strongly induced the death of C8-D1A astrocytes (Figure 1A,B). Proliferation of cells treated for four days was also evaluated (Appendix A). Treatment with axitinib or etoposide was sufficient to reduce proliferation of MB cells by more than 80% after 120 h. Proliferation of normal cells was also impaired in these conditions, although to a lower extent with axitinib as compared to etoposide (Appendix A). This result was in favor of a cytostatic rather than cytotoxic effect of axitinib towards nontumor cells.

To get insight into the dynamics of the effect of axitinib and etoposide treatments, long-term proliferation was assessed in a cumulative population doubling (CPD) assay conducted on two MB cell lines (DAOY and HD-MB03) and two normal cell lines (C8-D1A and HDF) (Figure 1C). Continuous treatment of DAOY and HD-MB03 cells with either axitinib or etoposide strongly impaired their proliferation (Figure 1C). Continuous treatment of nontumor cells C8-D1A and HDF also impaired their proliferation, although axitinib had a milder effect on C8-D1A cells compared to the reference treatment. Importantly, normal cells were able to proliferate again after the removal of axitinib (Figure 1C). This result suggests a cytostatic rather than a cytotoxic effect of this treatment on normal cells and further supports the selective inhibitory effect of axitinib on tumor versus nontumor cells. Considering our results and previous studies showing antiproliferative effects of axitinib on MB cells [33,34] and no efficiency of sunitinib on pediatric brain tumors [19], we focused our study on axitinib.

We generated spheroids with five different MB cell lines originally isolated form Shh (DAOY), group 3 (HD-MB03, D283, D458) and group 4 (CHLA-01-MED) tumors. Axitinib abolished 3D growth of MB cells in a way comparable to the etoposide reference treatment (Figure 1D and Appendix A). A combination of axitinib/etoposide (5 µM and 1 µM, respectively) reduced the growth of DAOY and HD-MB03 spheroids more efficiently than etoposide alone (Figure 1D). These results suggest that axitinib may be combined with the reference chemotherapy with etoposide.

### 3.2. Axitinib Leads to Low Toxicity towards Normal Brain Cells

To assess the relative selectivity of the treatments towards normal and tumor cells in a single assay, we set up cocultures of fluorescently labelled HD-MB03, DAOY and C8-D1A cells (Figure 2). MB cells stably expressing RFP and C8-D1A astrocytes stained with a neuro-DiO probe were mixed and treated with axitinib or etoposide. Both treatments resulted in a 40–60% enrichment of the proportion of astrocytes relative to the total number of cells (Figure 2A,B). We performed the same experiment in 3D mixed cultures of C8-D1A astrocytes and HD-MB03 cancer cells (Figure 2C–E). Both treatments resulted in a strong decrease in the size of spheroids and a decrease in the relative fluorescence of MB vs. normal cells (Figure 2D,E). These results confirm that axitinib and etoposide selectively impair tumor cell survival.

Two concerns regarding the use of a new compound to treat pediatric brain tumors are as follows: (1) the toxicity of the compound towards developing organisms and (2) its ability to cross the blood–brain barrier (BBB). Twenty-day-old weaned rats were treated daily with 50 mg/kg axitinib administered by oral gavage for four weeks. Although no direct correlation can be established between rat and human age, this early stage has been proposed to be comparable with a period of around one year of age for humans [35]. Axitinib had no effect on the general behavior and growth rate of both female and male animals (Figure 3A). At the end of the experiment, the animals were sacrificed and the axitinib content of their cerebella was measured by means of liquid chromatography coupled to mass spectrometry (LC/MS). Importantly, we showed that axitinib was present in the cerebellum (Figure 3B). Although a previous report showed that axitinib was flushed out of the brain very efficiently by ABCG2 and ABCB1/2 efflux pumps after acute exposure [36,37], there were no data previously available in the literature regarding the effect of a chronic treatment on brain axitinib accumulation. Thus, to get further insight into the mechanism leading to brain axitinib accumulation, we hypothesized that BBB permeabilization may be a result of long-term treatment with axitinib. To test this hypothesis, we used an in vitro model of BBB [29]. We first determined the toxicity of three doses of axitinib (1, 5 and 25 µM) on human brain and umbilical vein endothelial cells (hereinafter abbreviated as BEC and HUVEC, respectively) (Figure 3C). Axitinib only displayed toxicity in brain endothelial cells after three days at doses higher than 1 µM. We therefore chose the doses of 0.5, 1 and 5 µM and a 3-day-long treatment to test the effect of axitinib on the permeability of our in vitro BBB model (Figure 3D). We showed that axitinib permeabilized the barrier formed by BEC or HUVEC at nontoxic doses. These results support the idea that axitinib induces permeabilization of the BBB, allowing it to accumulate in the brain.

### 3.3. Axitinib and Axitinib/Etoposide Combination Decrease Tumor Growth

We next performed tumor xenograft experiments to address the efficiency of axitinib in MB. Importantly, receptors and kinases targeted by axitinib are expressed in human MB tumors (Appendix A). We used cell lines that express high levels of VEGF to generate tumors (Appendix A). HD-MB03 spheroids were implanted in the mouse cerebella and intracranial tumor engraftment and growth were monitored by means of BLI (Figure 4 and Appendix A). Tumors were considered engrafted when two consecutive increases in luciferase activity were measured. At that point the animals were randomized, and the treatments started (Appendix A). The treatments consisted in either metronomic administration of high doses of axitinib or high doses of etoposide (50 mg/kg and 30 mg/kg, respectively, administered orally five times a week) [38,39]. To test the possibility of compensating etoposide dose reduction with axitinib in a future clinical trial, we treated the animals with a combination of half the doses of axitinib and etoposide (25 mg/kg and 15 mg/kg, respectively, administered orally five times a week). Both metronomic axitinib and combined axitinib/etoposide treatments increased survival of the animals and reduced tumor growth while metronomic etoposide treatment had no significant effect on survival (Figure 4A,B and Appendix A). The efficiency of the axitinib/etoposide combination was confirmed in an independent experiment (Appendix A). Histological analysis of the tumors revealed that all the treatments decreased the number of proliferative Ki67-positive cells (Figure 4C,D). Metronomic axitinib, etoposide and axitinib/etoposide treatments also decreased the number of endothelial cell (CD31-positive) and pericyte (αSMA-positive) double-stained structures representative of vessels covered with pericytes (Figure 4E,F). These results suggest that all the treatments administered in a metronomic schedule directly impact the proliferation of cancer cells and neoangiogenesis of intracranial tumors. However, axitinib and axitinib/etoposide combination only increased the mouse survival (Figure 4 and Appendix A). Hence, our results suggest that metronomic administration of axitinib alone or a combination of lower doses of etoposide and axitinib directly impacted proliferation of cancer cells and neoangiogenesis of intracranial tumors. These results also suggest that the treatments can cross the blood–brain barrier (BBB).

We confirmed these results in experimental MB generated with HD-MB03 or DAOY cells xenografted subcutaneously. To some extent, it can be considered as a suitable metastatic model. High-dose axitinib (50 mg/kg administered orally thrice a week), high-dose etoposide (30 mg/kg administered orally thrice a week) or a combination of half the doses of axitinib and etoposide (25 mg/kg and 15 mg/kg, respectively, administered orally thrice a week) were used. The median time to reach the endpoint volume (1000 mm^3^) for the control, axitinib and etoposide-treated subcutaneous HD-MB03 tumors was 21 days (range, 11–39 days), 18 days (range, 13–63 days) and 18 days (range, 11–60 days), respectively (Appendix A). Instead, the tumors treated with the axitinib/etoposide combination reached the same size after a median time of 43 days (range, 15–70 days) (Appendix A). As tumor growth was biphasic, with “lag time” followed by a rapid growth, we applied a linear regression method to analyze the growth rates in the actual growth phase (Appendix A). Tumor growth rates were reduced in the mice treated with either axitinib alone or in combination with etoposide (Figure 5A). Survival analysis was performed based on the proportion of mice reaching the endpoint measurement of the tumor volume over time (Figure 5B). Only the axitinib/etoposide combination prolonged the mouse survival (*p* = 0.029). All the treatments reduced the proportion of Ki67-positive proliferative cells within HD-MB03 tumors (Figure 5C,D). Axitinib alone or combined with etoposide increased the surface of necrotic areas in subcutaneous HD-MB03 tumors (Figure 5C,E). In these tumors, axitinib or etoposide did not impact the number of endothelial cells (CD31-positive) and/or of pericytes/cancer-associated fibroblasts (αSMA-positive) (Figure 5C,F). Instead, the axitinib/etoposide combination decreased the number of αSMA- and/or CD31-positive cells and increased the number of double-stained αSMA/CD31 structures representative of arterioles (Figure 5C,G). These results suggest that the combination treatment is specifically responsible for the restoration of functional arterioles within subcutaneous HD-MB03 tumors allowing the chemotherapeutic agent to reach tumor cells more efficiently [39].

We also performed a subcutaneous xenograft experiment with the Shh group’s DAOY cells (Figure 6 and Appendix A). The lag time preceding tumor growth was more homogenous as compared to that of HD-MB03 cells, ranging approximately from 35 to 45 days (Appendix A). Mathematical modelling suggested that axitinib did not impair tumor growth while etoposide and axitinib/etoposide decreased tumor growth rates (Figure 6A). Histological analysis of the tumors showed that all the treatments decreased cell proliferation and increased fibrosis (Figure 6C–E) without significantly affecting the blood vessels in this model. These results suggest that axitinib mediates its antitumoral effect mainly by decreasing tumor cell proliferation in all the models we tested, while its antiangiogenic effect varies from one model to another. 

### 3.4. High Expression of Axitinib Targets Is Linked to Poor Prognosis in Shh Patients

To define the genetic subgroup of the patients that may benefit from axitinib, we correlated the expression of axitinib targets (VEGF receptors 1, 2, 3, cKit and PDGFRA and PDGFRB) to survival in available sets across 763 primary samples of MB generated by Cavalli et al. (Table 4) [40]. For group 3 patients, overexpression of cKit and PDGFRA was associated with a shorter survival (*p* = 0.04 and *p* = 0.012, respectively) while overexpression of PDGFRB and VEGFR1 correlated with a longer survival (*p* = 0.017 and *p* = 0.041, respectively). For group 4 patients, overexpression of PDGFRB and VEGFR1 is synonymous to shorter survival (*p* = 0.04 and *p* = 5.5 × 10^−3^, respectively) while it is the contrary for VEGFR2 (*p* = 0.043). For Shh patients, rapid death correlated with overexpression of cKit, PDGFRA, VEGFR1 and VEGFR2 (*p* = 8.7 × 10^−5^, *p* = 6.7 × 10^−5^, *p* = 4.8 × 10^−3^, *p* = 5.2 × 10^−3^, respectively) while overexpression of PDGFRB and VEGFR3 correlated with a longer survival (*p* = 5 × 10^−8^, *p* = 0.015, respectively). We attributed a relative strength to each parameter, and we gave a “weight” of +2 to good prognostic markers and a “weight” of −2 to bad prognostic markers. The worst score was obtained for patients of the Shh subgroup (−5). Therefore, these patients would be good responders to axitinib. These in silico results are not in agreement with those obtained with the xenografts experiments (axitinib is more efficient in a model of group 3 tumors as compared to a model of Shh tumors). These results strongly suggest that determination of the expression of axitinib targets in the primary tumor or at relapse on reference treatment is recommended before administration of the compound. 

## 4. Discussion

The aim of our study was to determine the relevance of the last-generation antiangiogenic compounds for the treatment of MB with limited side effects, especially in the pediatric context. In this study, we showed that some antiangiogenic TKi efficiently kill tumor cells initially isolated from any MB molecular subgroups. Amongst these compounds, axitinib displays a pleiotropic effect on MB cells from any molecular subgroups and is capable of permeabilizing (in vitro) and crossing (in vivo) the BBB. We further showed that axitinib was the compound displaying the lowest toxicity towards noncancer cells, suggesting it may induce lower toxicity in pediatric patients. Consistently, daily high-dose treatment of juvenile rats did not lead to acute toxicity. Finally, axitinib alone or combined with a reduced dose of etoposide is efficient against MB in vivo both in subcutaneous and cerebral tumors. Our decision to evaluate in vitro and in vivo the effect of the etoposide + axitinib combination was taken based on a clinical trial proposal that was designed during the course of the experiments. Furthermore, a phase I clinical study has recently established the pharmacokinetics and the maximum tolerated and recommended dose of axitinib in a pediatric context [20]. This study also pinpoints the preliminary evidence of axitinib efficacy in children, although no patient presenting with a brain tumor was included in this trial. Treatment with axitinib alone or combined with the PI3K inhibitor GDC-0941 has also shown efficacy in several in vitro MB models [34,35]. Our work and those studies pave the way towards the use of axitinib to treat pediatric patients with solid tumors of the central nervous system.

We attempted to find the axitinib targets that were inhibited upon exposure (Appendix A). However, we were unable to identify a mechanism responsible for its toxicity towards MB cells. Indeed, axitinib impairs a wide variety of kinase and non-kinase targets [41,42]. Thus, the failure in identifying the mechanism responsible for axitinib toxicity towards MB cells probably depends on combined impairment of several targets. The ability to target many different proteins may also explain the efficacy of axitinib in MB cells and tumors belonging to different molecular subgroups. Moreover, despite the most up-to-date WHO classification of MB considers molecular profiling, applying molecular methods to each case of MB in the world is not a realistic option. Therefore, treatments effective towards any form of MB and not specific to a given molecular subgroup still need to be seriously considered.

Crossing the BBB to treat cerebral tumors is extremely challenging. Indeed, the BBB is formed by the walls of brain capillaries and strictly controls the components that can enter the brain [43,44]. Endothelial cells of the BBB form a highly impermeable cell layer and express high amounts of efflux pumps that are in charge of exporting molecules out of the brain. Axitinib is actively removed from the cerebral compartment by the action of the ABCB1 efflux pump system in mice treated with a single dose [37,45]. Nonetheless, axitinib has also proven efficient against orthotopically implanted mouse glioblastoma [46], supporting the idea that it can access brain tumors. Importantly, the formation and maintenance of the BBB are under the control of the Wnt/β-catenin pathway [47,48,49]. Axitinib efficiently inhibits the activity of this pathway [42]. In this study, we also evidenced that it permeabilizes an in vitro BBB model and that it can be found in the brain of chronically treated rats. Therefore, axitinib could permeabilize the BBB through inactivation of the Wnt/β-catenin pathway. This study describes the inhibition of brain experimental tumors by an axitinib/etoposide combination which can be related to permeabilization of the BBB by axitinib. The combination treatment probably induced a better accessibility of the chemotherapeutic agent to tumor cells through normalization of the intratumor blood vessels, a concept already proposed as a mechanism of action of antiangiogenic drugs [50].

## 5. Conclusions

We showed the relevance of axitinib in a pediatric context together with its predicted low toxicity and ability to target heterogeneous tumors in the brain. Hence, it represents a promising compound to add in the therapeutic arsenal against pediatric brain tumors.

## Figures and Tables

**Figure 1 cancers-14-00070-f001:**
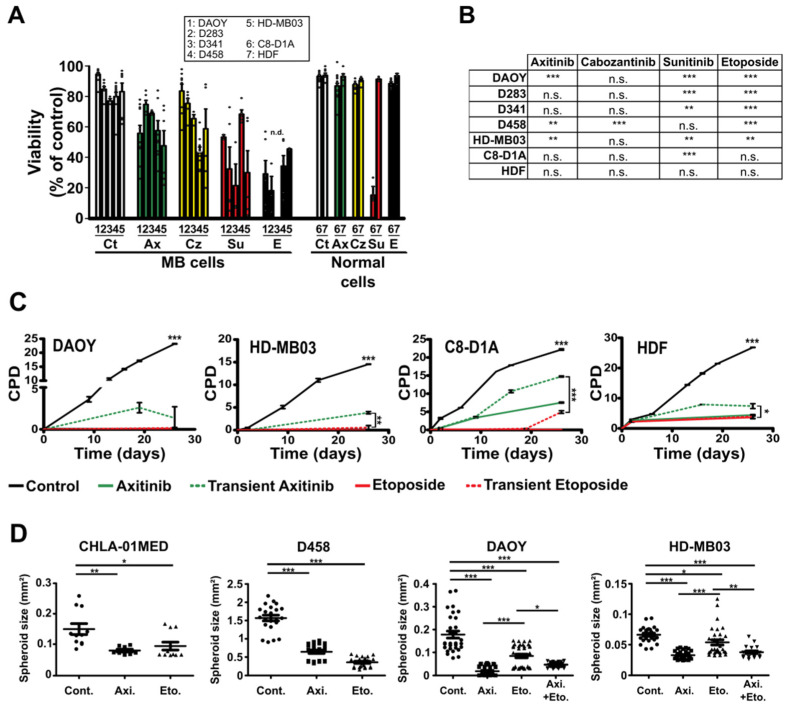
Antiangiogenic compounds strongly impair the viability and proliferation of MB cells. (**A**) Viability of the indicated cell lines treated for 48 h with axitinib (5 µM, Ax, green), cabozantinib (5 µM, Cz, yellow), sunitinib (5 µM, Su, red) or etoposide (1 µM, E, black). All the compounds were dissolved in DMSO, the amount of which was adjusted to be the same in every condition. Control conditions (Ct, white) were also treated with the same amount of DMSO. Histograms represent the means ± SEM; each independent datapoint is represented (white circles) (n.d.: not determined). (**B**) Table showing the statistical significance of the viability results for each compound compared to the control condition (*** *p* < 0.001, ** *p* < 0.01, n.s.: nonsignificant, Student’s *t*-test comparing at least three independent experiments). (**C**) Cumulative population doubling (CPD) of the indicated MB and normal cell lines continuously treated with 5 µM axitinib (continuous green lines), 1 µM etoposide (continuous red lines) or transiently treated for 48 h with the same compound concentrations (dotted lines) (datapoints represent the means ± SEM of a representative experiment; *** *p* < 0.001, ** *p* < 0.01, * *p* < 0.05, one-way ANOVA test, the results are statistically nonsignificant unless otherwise stated). (**D**) Dot plots showing the endpoint size measurements of the spheroids generated with the indicated MB cell lines and continuously treated with 5 µM axitinib (Axi.), 1 µM etoposide (Eto) (CHLA-01MED and D458) or a combination of axitinib (5 µM) and etoposide (1 µM; Axi. + Eto.). The controls were all treated with a concentration of DMSO corresponding to the one used as a vehicle for the drugs (data collected from at least three independent experiments are presented; the bars represent the means ± SEM; *** *p* < 0.001, ** *p* < 0.01, * *p* < 0.05, one-way ANOVA test, the results are statistically nonsignificant unless otherwise stated).

**Figure 2 cancers-14-00070-f002:**
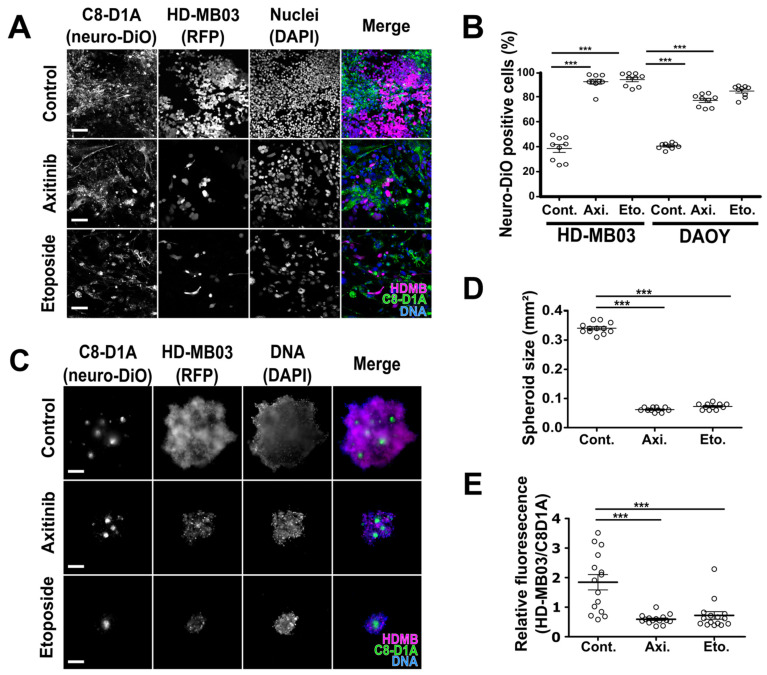
Axitinib selectively impacts proliferation of MB cells in 2D models. (**A**) Confocal images of neuro-DiO-stained primary astrocyte (C8-D1A, green) and RFP-expressing MB cell (HD-MB03, magenta) cocultures treated for 4 days with 5 µM axitinib (Axi.) or 1 µM etoposide (Eto.) (Hoechst 33342 nuclear DNA counterstain is shown in blue; scale bars: 100 µM). (**B**) FACS quantification of neuro-DiO-positive C8-D1A cells relative to the total number of cells according to the cocultured MB cell line (HD-MB03 or DAOY) and the treatment (Axi.: 5 µM axitinib; Eto.: 1 µM etoposide) (horizontal lines and error bars represent the means ± SEM and all individual datapoints of the three independent experiments are plotted; *** *p* < 0.001, one-way ANOVA test, the results are statistically nonsignificant unless otherwise stated). (**C**) Mixed spheroids generated with neuro-DiO-stained astrocytes (C8-D1A, green) and RFP-expressing MB cells (HD-MB03, magenta) treated for 7 days with 5 µM axitinib or 1 µM etoposide (scale bars: 250 µM). (**D**) Dot plots showing the endpoint size measurements of the C8-D1A/HD-MB03 mixed spheroids treated with the indicated compounds. (**E**) Dot plots showing the relative fluorescence of HD-MB03 and C8-D1A cells (data collected from at least three independent experiments are presented; the bars represent the means ± SEM; *** *p* < 0.001, one-way ANOVA test, the results are statistically nonsignificant unless otherwise stated).

**Figure 3 cancers-14-00070-f003:**
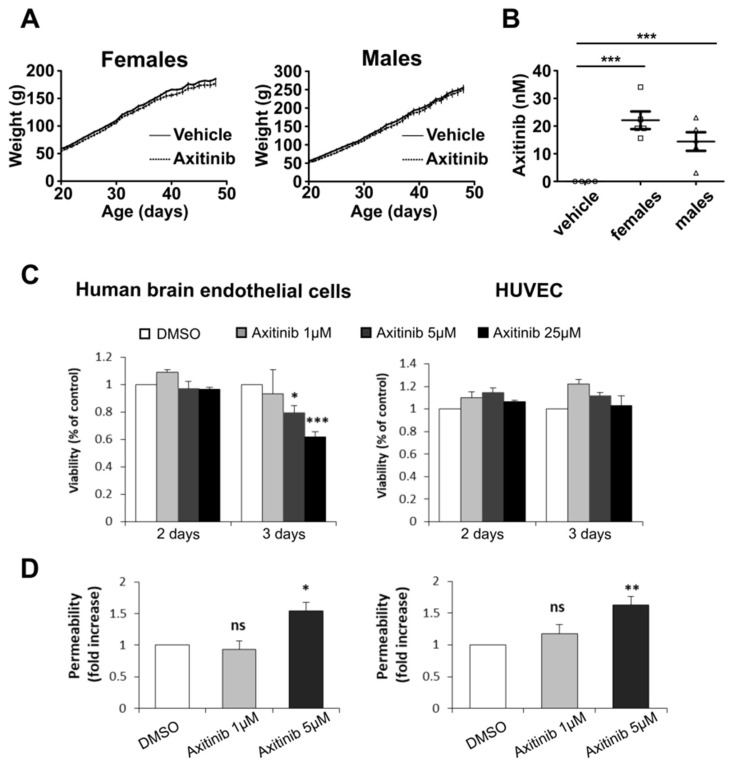
Axitinib shows no major chronic toxicity and can pass the blood–brain barrier of juvenile mammals. (**A**) Growth curves of female and male 20-day-old Wistar rats treated with axitinib (50 mg/kg/day) for 28 days (datapoints represent the mean weight ± SEM, *n* = 5 rats per group). (**B**) Dot plots representing the measurements of axitinib concentrations in rat cerebella after 28 days of axitinib (50 mg/kg/day) or vehicle alone treatments (*n* = 5 cerebella per group, bars represent the means ± SEM). (**C**) MTT assay analysis of axitinib toxicity towards human brain endothelial (BEC) and human umbilical vein endothelial cells (HUVEC). (**D**) Relative permeability of human brain endothelial cells and HUVEC in an in vitro endothelium permeability assay. Note: * *p* < 0.05, ** *p* < 0.01, *** *p* < 0.001, one-way ANOVA test.

**Figure 4 cancers-14-00070-f004:**
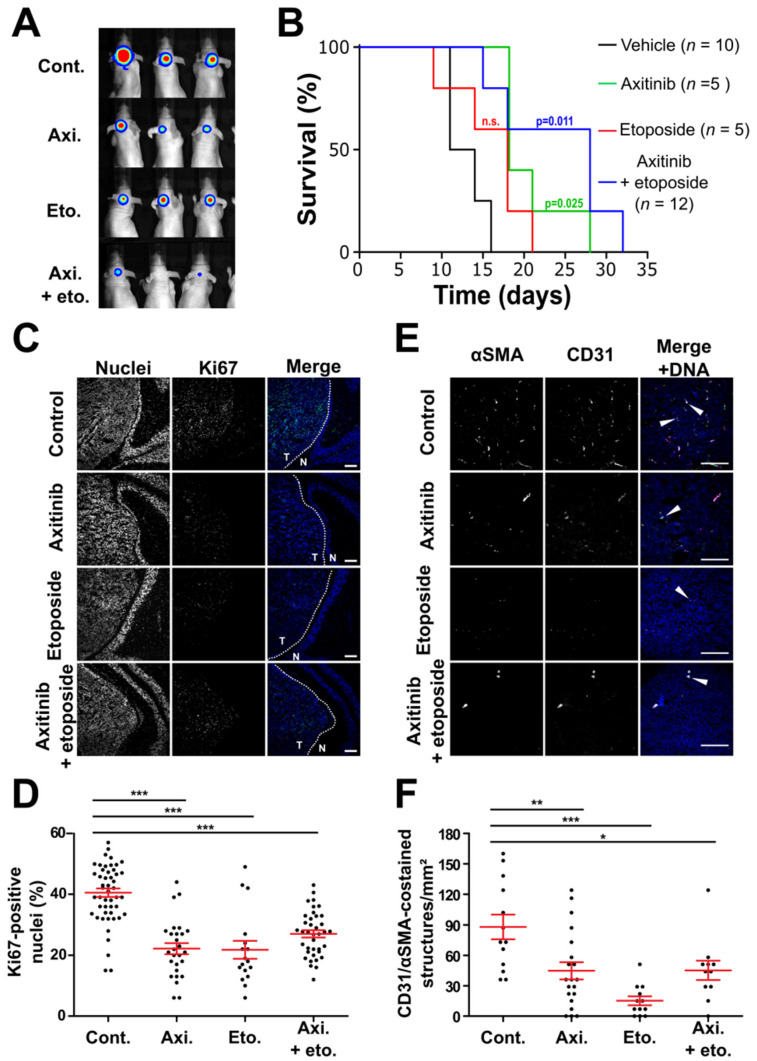
Axitinib and axitinib/etoposide combination reduce the HD-MB03 intracranial tumor growth rate and increase survival more efficiently than etoposide. (**A**) Representative picture of the BLI signal intensity after 11 days of treatment. (**B**) Survival curves presenting the proportion of alive mice as a function of time after tumor implantation (black: control treatment; green: axitinib treatment; red: etoposide treatment; blue: axitinib/etoposide combined treatment; the *p*-value of a logrank test comparing the control and the axitinib or axitinib/etoposide groups is indicated with a relevant color; all the other comparisons were not statistically significant; the total number of animals for each group is indicated in brackets). (**C**) Images of sections of HD-MB03 intracranial tumors treated with the indicated compounds. Proliferative cells are revealed by nuclear Ki67 immunofluorescent staining (green) and Hoechst33342 DNA counterstaining (blue). Dotted lines delineate the border between normal (N) and tumoral (T) tissues (scale bars: 200 µm). (**D**) Dot plot representing the quantification of the proportion of Ki67-positive nuclei (*** *p* < 0.001, one-way ANOVA test, at least four independent tumors were analyzed). (**E**) Images of sections of HD-MB03 orthotopic tumors treated with the indicated compounds. The vasculature is revealed by CD31 and αSMA immunofluorescent staining (green and magenta, respectively) and Hoechst33342 nuclear DNA counterstaining (blue) (scale bars: 100 µm). (**F**) Dot plot representing the number of functional blood vessels per mm² in the indicated experimental conditions (bars represent the means ± SEM; *** *p* < 0.001, ** *p* < 0.01, * *p* < 0.05, one-way ANOVA test, at least four independent tumors were analyzed).

**Figure 5 cancers-14-00070-f005:**
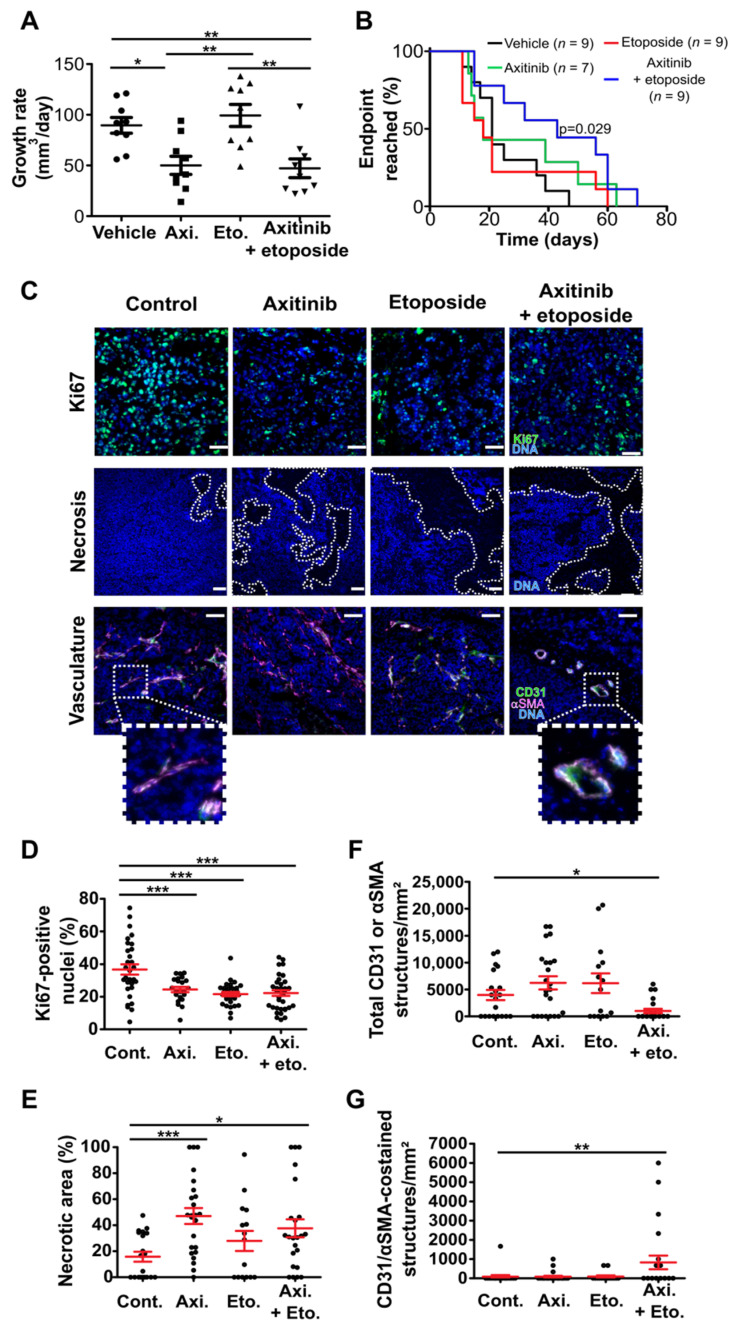
Axitinib and axitinib/etoposide combination treatments reduce subcutaneous HD-MB03 tumor growth and cell proliferation, induce necrosis and tumoral fibrosis and impair tumor vascularization. (**A**) Dot plots showing the growth rate of individual subcutaneous tumors treated with the indicated compounds (horizontal lines and error bars represent the means ± SEM; ** *p* < 0.01, * *p* < 0.05, one-way ANOVA test, the results are statistically nonsignificant unless otherwise stated). (**B**) Curves presenting the proportion of mice reaching the endpoint volume of the tumor as a function of time after tumor implantation (black: control treatment; green: axitinib treatment; red: etoposide treatment; blue: axitinib/etoposide combined treatment; the *p*-value of a logrank test comparing the control and axitinib/etoposide groups is indicated; all the other comparisons were not statistically significant; all the data presented were pooled from two strictly independent experiments and the total number of animals for each group is indicated in brackets). (**C**–**G**) Histological analysis of subcutaneous HD-MB03 xenografts. (**C**) Representative images of the sections of tumors treated with the indicated compounds: proliferative cells revealed by Ki67 immunofluorescent staining (green) and Hoechst33342 nuclear DNA counterstaining (blue); necrosis revealed by DNA counterstaining (blue) (dotted lines delineate necrotic areas); vasculature revealed by CD31 and αSMA immunofluorescent staining (green and magenta, respectively) and Hoechst33342 nuclear DNA counterstaining (blue) (images are representative of at least four independent tumors, scale bars: 100 µm). (**D**) Dot plot representing the quantification of the proportion of Ki67 positive nuclei in the indicated experimental conditions. (**E**) Dot plot representing the quantification of the proportion of necrotic area in the indicated experimental conditions, (**F**) the number of CD31 and/or αSMA structures per µm² and (**G**) the number of functional blood vessels per mm² (bars represent the means ± SEM; *** *p* < 0.001, ** *p* < 0.01, * *p* < 0.05, one-way ANOVA test).

**Figure 6 cancers-14-00070-f006:**
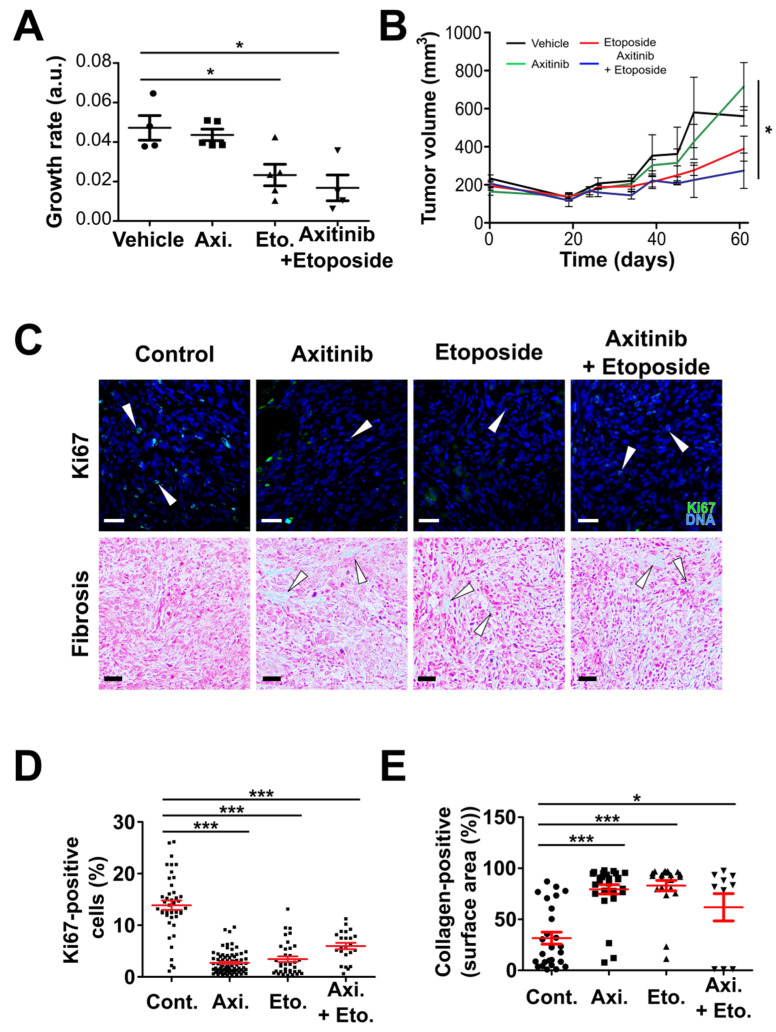
Etoposide and axitinib/etoposide combination reduce subcutaneous DAOY tumor growth and tumor cell proliferation and induce tumoral fibrosis. (**A**) Dot plots showing the growth rate of individual subcutaneous tumors treated with the indicated compounds (horizontal lines and errors bars represent the means ± SEM; * *p* < 0.05, one-way ANOVA test, the results are statistically nonsignificant unless otherwise stated). (**B**) Mean growth curves of subcutaneous DAOY tumors treated with the indicated compounds (errors bars represent the means ± SEM; * *p* < 0.05, one-way ANOVA test, the results are statistically nonsignificant unless otherwise stated). (**C**–**E**) Histological analysis of subcutaneous DAOY xenografts (* *p* < 0.05, Student’s *t*-test, the results are statistically nonsignificant unless stated otherwise). (**C**) Representative images of the sections of tumors treated with the indicated compounds. Proliferative cells were revealed by Ki67 immunofluorescent staining (green) and Hoechst33342 nuclear DNA counterstaining (blue) (white arrowheads: Ki67-positive nuclei). Tumoral fibrosis was revealed by Masson’s trichrome staining (nuclei: dark pink; cytoplasm: light pink; collagen: light blue; white arrowheads: collagen-positive areas; scale bars: 100 µM). (**D**) Dot plots representing the quantification of the proportion of Ki67-positive cells in the indicated experimental conditions and (**E**) the proportion of the image fields displaying consistent collagen staining (bars represent the means ± SEM; *** *p* < 0.001, * *p* < 0.05, one-way ANOVA test, the results are not statistically significant unless otherwise stated).

**Table 1 cancers-14-00070-t001:** Toxicity of antiangiogenic compounds and chemotherapeutic agents towards normal and MB cells. The IC50 (µM) values for various compounds determined by means of MTT tests after 48 h exposure of nontumor and MB cells (values are mean IC50 ± standard error of the mean from at least three independent experiments).

	Normal Cells		Medulloblastoma Cells			
IC50 (µM)	HDF	C8D1A	DAOY	HD-MB03	D458	CHLA-01-Med
Sunitinib	1.8 ± 0.04	5.7 ± 2.2	7.9 ± 5.5	4.6 ± 0.3	4.2 ± 0.3	7.5 ± 0.8
Cabozantinib	4.1 ± 1.9	4 ± 2.2	8.3 ± 1.2	0.78 ± 0.2	5.1 ± 1.6	11.4 ± 2.1
Carbo/Eto	1.6 ± 1.1	3 ± 0.8	1 ± 0.4	0.16 ± 0.07	0.56 ± 0.7	0.11 ± 0.08
Axitinib	>100	>100	2.3 ± 1.5	0.47 ± 0.2	0.49 ± 0.4	3.9 ± 0.23

**Table 2 cancers-14-00070-t002:** Specificity indexes for three antiangiogenic compounds and reference chemotherapy in normal and MB cells. Specificity indexes were determined by calculating the ratio between the IC50 of a given drug for either human dermal fibroblasts (HDF) or murine primary astrocytes (C8-D1A) and the IC50 of the same drug for each of the MB cell lines (D458, DAOY, HD-MB03 and CHAL-01-Med).

Specificity Index	Relative to HDF				Relative to C8D1A			
	DAOY	HD-MB03	D458	CHLA-01-Med	DAOY	HDMB	D458	CHLA-01-Med
Sunitinib	0.23	0.39	0.43	0.24	0.72	1.24	1.36	0.76
Cabozantinib	0.49	5.26	0.80	0.36	0.48	5.13	0.78	0.35
Carbo/Eto	1.60	10.00	2.86	14.3	3.00	18.75	5.36	26.8
Axitinib	>40	>200	>200	>25	>40	>200	>200	>25

**Table 3 cancers-14-00070-t003:** Comparison of the toxicity of etoposide, carboplatin and etoposide/carboplatin combination treatments. The IC50 (µM) values determined by means of MTT tests after 48 h exposure of three MB cell lines are presented (values are mean IC50 ± standard error of the mean from at least three independent experiments).

IC50 (µM)	DAOY	HD-MB03	D458	HDF	C8-D1A
Etoposide	0.2	0.09 ± 0.006	0.52 ± 0.15	0.06 ± 2	2.2 ± 0.6
Carboplatin	>20	>20	>20	>20	>20
Etoposide/carboplatin	1 ± 0.41	0.17 ± 0.05	0.58 ± 0.51	2 ± 1.1	3 ± 0.8

**Table 4 cancers-14-00070-t004:** Scoring of the association between the axitinib target expression levels and the patients’ survival rates.

Gene/Best Cutoff	Group 3 (113 Patients)	Group 4 (264 Patients)	Shh (172 Patients)
cKit	*p* = 0.04	*p* = 0.083	*p* = 8.7 × 10^−^^5^
PDGFRA	*p* = 0.012	*p* = 0.061	*p* = 6.7 × 10^−^^5^
PDGFRB	*p* = 0.017	*p* = 0.04	*p* = 5 × 10^−^^8^
VEGFR1	*p* = 0.041	*p* = 5.5 × 10^−^^3^	*p* = 4.8 × 10^−^^3^
VEGFR2	*p* = 0.325	*p* = 0.043	*p* = 5.2 × 10^−^^3^
VEGFR3	*p* = 0.075	*p* = 0.1	*p* = 0.015
Score	0	−2	−5

## Data Availability

The pipeline used for automated image analysis of Ki67-positive nuclei with the CellProfiler 3.0 software [28] is available at https://github.com/TTteam-CSM/image_analysis, accessed on 2 December 2021. The code used for tumor growth statistical analysis is available upon request to V.P-B. All the datasets shown in the figures are available from the corresponding author on reasonable request.

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
