# Peer review of "Antiangiogenic Compound Axitinib Demonstrates Low Toxicity and Antitumoral Effects against Medulloblastoma"

_cancers, 2021, doi:10.3390/cancers14010070_

Round 1

Reviewer 1 Report

The aim of the study was to find efficient anticancer therapies with less toxic effects for the treatment of medulloblastoma. Therefore, the authors explored the effect of three anti-angiogenic compounds (axitinib, sunitinib, cabozanitinib) in vitro and in vivo. The study is interested and in general well conducted. However, I have several comments:

  • The authors write about antiangiogenic effects, however, the three drugs which are tested in vitro and in vivo do exert antineoplastic effects by inibiting several tyorsine kinases such as PDGFRA/B or c-KIT, etc. This is also stated in the first sentence of the results part and might attribute to most (not all) of their findings. Therefore, the abstract and title might be misleading.
  • Please state the three drugs in the abstract (p.1 line 38)
  • The sentences in the lines 148 and 153 are redundant.
  • Please visually separate normal cells and medulloblastoma cells for better overview in table 1/2/3.  
  • Figure 1C: What does continuously treated with the respective drug mean? And why are there in general less measurements in the treatment groups compared to the control group?
  • Figure 3C/D: Please state the dosages more clearly: e.g.: instead of 1 axi: Axitinib 1 µM.
  • Please state the number of animals used for each group directly in the figure (Fig 4B and Fig 5B).
  • How does the applied dosage of axitinib 50 mg/kg/day in the animal experiments correspond to current human dosages?
  • A dose-response matrix of axitinib and etoposide in several cell lines would be a great addition to see the synergistic effects.
  • Discussion: line 596: Molecular profiling of MBs is already integrated in the WHO classification.

Author Response

Dear Reviewer,

We are pleased to provide you with responses to your concerns about our manuscript (please see attachment). We thank you for your extremely relevant remarks that allowed a substancial improvement of the manuscript.

Best regards,

Vincent Picco

Reviewer 2 Report

In this study, Marina Pagnuzzi-Boncompagni et al. studied three different molecules with anti-angiogenic properties against medulloblastoma (MB) and demonstrated that the compound axitinib was able to reduce cell viability in five medulloblastoma different cell lines without significant effects on normal cells. Moreover, axitinib showed the ability to enter the brain compartment exerting anti-tumoral effects in vivo.

The analyzed topic is of relative interest and the presented data are quite convincing, that if the Authors could address the points below reported, the paper might be proposed for publication in Cancers

Section Abstract:

The authors should reduce the Background in favour the aim of the study, Methods and Results. In the Methods the subgrouping of cell types should be reported and in the results the effects of the other compounds and the combination therapies should at least be mentioned.

Section Introduction:

Line 77: The Authors should better clarify to the reader the mechanisms of action of the indicated therapies. In addition, they are only valid for a subgroup of MB. Why are only these indicated?

Line 99: The Authors should indicate the cell models used in the study and justify their choice.

Authors should provide explanations.

Section Materials and Methods:

Paragraph 2.1. Cell lines In this paragraph the species of each cell line is not reported, the Authors should insert this information.

Paragraph 2.3. MTT, proliferation assays cell viability and cumulative population doubling assays Authors should explain why they use different concentrations in normal cells than those used in MB cell lines

The methods using the cytometer must be better described, the purposes for which the methods are carried out, shall be reported at least in the methods.

 Section Results:

The results in general are many and often poorly described, making too many references to the data reported in the supplementary figures, distracting the reader in focusing on the results.

The Authors should clearly indicate the species of MB cell lines and why they use as normal cell lines the murine primary astrocytes and the human dermal fibroblasts. The Authors use the Selectivity index, based on the ratio of the IC50 in normal cells and the IC50 in MB cell lines, but which normal cells do they consider? The Authors should insert the references about this parameter. There are reference values for the selectivity index? for example what does the ratio equal to 1 mean?

In the sentence: “We showed that 303 sunitinib, axitinib and cabozantinib impact group 3 (D283, D458 and HD-MB03), group 4 304 (CHLA-01-MED) and a putative Shh group (DAOY) MB cell lines [4] in a way comparable to the reference treatment (Table 1 and Supplementary Fig. S1)” the Authors observe a comparable effect to the reference treatment, but this assertion is not clear, Authors should provide explanations. In table 1 SEM is not reported for CHLA-01-Med. The data on cell viability in figure 1A are confusing and unclear. What are the times of the treatments in Figure 1A? 4 days, 48 h?  Have the treatments been added again? What does the term continuous mean? The Authors should provide explanations.

Why do the authors evaluate only the combination etoposide+axitinib?

The Authors should explain the mean of the results described in this phrase: In these tumors, axitinib or etoposide did not impact the number of endothelial cells (CD31 positive) and/or pericytes/cancer associated fibroblasts (αSMA positive) (Figure 5C, F). Instead, the axitinib/etoposide combination decreased the number of αSMA and/or CD31 positive cells and increased the number of double-stained αSMA/CD31 structures representative of arterioles (Figure 5 C, G).

Section Discussion:

In the Discussion, the authors should more clearly justify the results and help the reader to follow the logical reasoning behind the obtained results.

Author Response

Dear Reviewer,

We are pleased to provide you with the responses to your concerns about our manuscript (please see attachment). We thank you for your comments that allowed a very substancial improvement of the manuscript.

Best regards,

Vincent Picco

Round 2

Reviewer 1 Report

The authors addressed most of my previous concerns.

Author Response

Dear reviewer,

The authors would like to thank you for your help with improving the manuscript.

Best Regards,

Vincent Picco

Reviewer 2 Report

In this study, Marina Pagnuzzi-Boncompagni et al. studied three different molecules with anti-angiogenic properties against medulloblastoma (MB) and demonstrated that the compound axitinib was able to reduce cell viability in five medulloblastoma different cell lines without significant effects on normal cells. Moreover, axitinib showed the ability to enter the brain compartment exerting anti-tumoral effects in vivo.

The analyzed topic is of relative interest and the presented data are quite convincing, that if the Authors could address the points below reported, the paper might be proposed for publication in Cancers

Section Abstract:

The authors should reduce the Background in favour the aim of the study:

The response of the Authors is adequate

Methods and Results. In the Methods the subgrouping of cell types should be reported:

The response of the Authors is adequate

and in the results the effects of the other compounds and the combination therapies should at least be mentioned: The response of the Authors is not adequate, I simply asked to report in the results of the abstract the obtained data with the other drugs studied and combinations.

Section Introduction:

Line 77: The Authors should better clarify to the reader the mechanisms of action of the indicated therapies. In addition, they are only valid for a subgroup of MB. Why are only these indicated?

The response of the Authors is adequate.

Line 99: The Authors should indicate the cell models used in the study and justify their choice.

Authors should provide explanations.

The response of the Authors is adequate.

Section Materials and Methods:

Paragraph 2.1. Cell lines In this paragraph the species of each cell line is not reported, the Authors should insert this information.

The response of the Authors is acceptable.

Paragraph 2.3. MTT, proliferation assays cell viability and cumulative population doubling assays Authors should explain why they use different concentrations in normal cells than those used in MB cell lines

The response of the Authors is not adequate or fully sufficient. There was a misunderstanding. I asked to explain why different drug concentrations were used in normal cells compared to those used in cancer cells.

The methods using the cytometer must be better described, the purposes for which the methods are carried out, shall be reported at least in the methods.

The response of the Authors is acceptable.

Section Results:

The results in general are many and often poorly described, making too many references to the data reported in the supplementary figures, distracting the reader in focusing on the results.

The authors have slightly improved their manuscript on this observation.

The Authors should clearly indicate the species of MB cell lines and why they use as normal cell lines the murine primary astrocytes and the human dermal fibroblasts.

The response of the Authors is sufficient.

The Authors use the Selectivity index, based on the ratio of the IC50 in normal cells and the IC50 in MB cell lines, but which normal cells do they consider? The Authors should insert the references about this parameter. There are reference values for the selectivity index? for example what does the ratio equal to 1 mean?

The response of the Authors is not fully sufficient. The Authors should improve the description of the results by inserting what a low or high SI value means.

In the sentence: “We showed that 303 sunitinib, axitinib and cabozantinib impact group 3 (D283, D458 and HD-MB03), group 4 304 (CHLA-01-MED) and a putative Shh group (DAOY) MB cell lines [4] in a way comparable to the reference treatment (Table 1 and Supplementary Fig. S1)” the Authors observe a comparable effect to the reference treatment, but this assertion is not clear, Authors should provide explanations.

The response of the Authors is acceptable.

In table 1 SEM is not reported for CHLA-01-Med.

The response of the Authors is acceptable.

The data on cell viability in figure 1A are confusing and unclear. What are the times of the treatments in Figure 1A? 4 days, 48 h?  Have the treatments been added again? What does the term continuous mean? The Authors should provide explanations.

The response of the Authors is not sufficient and unclear. The text remains confused.

Why do the authors evaluate only the combination etoposide+axitinib?

The Authors should report their decision at least in the discussion

The Authors should explain the mean of the results described in this phrase: In these tumors, axitinib or etoposide did not impact the number of endothelial cells (CD31 positive) and/or pericytes/cancer associated fibroblasts (αSMA positive) (Figure 5C, F). Instead, the axitinib/etoposide combination decreased the number of αSMA and/or CD31 positive cells and increased the number of double-stained αSMA/CD31 structures representative of arterioles (Figure 5 C, G).

The response of the Authors is acceptable.

Section Discussion:

In the Discussion, the authors should more clearly justify the results and help the reader to follow the logical reasoning behind the obtained results.

The response of the Authors is acceptable.
